# Interaction between PHB2 and Enterovirus A71 VP1 Induces Autophagy and Affects EV-A71 Infection

**DOI:** 10.3390/v12040414

**Published:** 2020-04-08

**Authors:** Weitao Su, Shan Huang, Huimin Zhu, Bao Zhang, Xianbo Wu

**Affiliations:** Department of Epidemiology, School of Public Health, Southern Medical University, Guangzhou 510515, China; swt937872090@163.com (W.S.); 15625985968@163.com (S.H.); gg3612@163.com (H.Z.)

**Keywords:** enterovirus A71, VP1, prohibitin 2, autophagy, autolysosome, virus infection, virus replication

## Abstract

Enterovirus A71 (EV-A71) is a major pathogen that causes severe and fatal cases of hand-foot-and-mouth disease (HFMD). HFMD caused by EV-A71 seriously endangers children’s health. Although autophagy is an important antiviral defense mechanism, some viruses have evolved strategies to utilize autophagy to promote self-replication. EV-A71 can utilize autophagy vesicles as replication scaffolds, indicating that EV-A71 infection is closely related to its autophagy induction mechanism. VP1, a structural protein of EV-A71, has been reported to induce autophagy, but the underlying mechanism is still unclear. In this study, we found that the C-terminus (aa 251–297) of VP1 induces autophagy. Mass spectrometry analysis suggested that prohibitin 2 (PHB2) interacts with the C-terminus of the EV-A71 VP1 protein, and this was further verified by coimmunoprecipitation assays. After PHB2 knockdown, EV-A71 replication, viral particle release, and viral protein synthesis were reduced, and autophagy was inhibited. The results suggest that PHB2 interaction with VP1 is essential for induction of autophagy and the infectivity of EV-A71. Furthermore, we confirmed that EV-A71 induced complete autophagy that required autolysosomal acidification, thus affecting EV-A71 infection. In summary, this study revealed that the host protein PHB2 is involved in an autophagy mechanism during EV-A71 infection.

## 1. Introduction

Outbreaks of hand-foot-and-mouth disease (HFMD) have recently occurred to varying degrees worldwide, especially in the Asia-Pacific region [1,2]. Indeed, HFMD has become a global infectious disease that affects the health of children under five years of age [3,4]. Enterovirus A71 (EV-A71) is the main pathogen of HFMD and is also the main cause of severe and fatal HFMD cases. HFMD caused by EV-A71 can develop into a severe neurological disease, causing brainstem encephalitis, acute flaccid paralysis, pulmonary edema, and cardiopulmonary failure, with rapid progression and a high mortality rate [5]. Overall, the mechanisms of EV-A71 infection need clarification. 

Autophagy is a highly conserved degradative process in eukaryotes that can be divided into different stages: induction and formation of autophagosome precursors, extended expansion of autophagosomes, fusion with lysosomes, and degradation [6,7]. Autophagy is measured by evaluating autophagy flux, including the formation and degradation of autophagosomes. Autophagosomes are double-membrane vesicles that mediate cargo sequestration and delivery to lysosomes [8]. Some viruses use autophagosomes as their site of replication. Many RNA viruses, such as poliovirus (PV), coxsackievirus B3 (CVB3), Japanese encephalitis virus (JEV), and hepatitis C virus (HCV), can induce and utilize autophagy to promote self-replication [9,10,11,12,13]. Increased amounts of autophagosomes, as well as colocalization of the autophagy marker protein and viral proteins, have been observed in cells infected by these viruses, and inhibition of autophagy reduces viral titers [13]. Studies have demonstrated that EV-A71-induced autophagy promotes viral replication in vivo and in vitro, suggesting that autophagy vesicles can serve as replication scaffolds for EV-A71 [14,15]. EV-A71 belongs to the enterovirus A species of the Picornaviridae family and contains a single-stranded positive-sense RNA. The coding subregions of EV-A71 are as follows: P1, which encodes four structural proteins (VP1–VP4); and P2 and P3, which encode seven nonstructural proteins (2A–2C and 3A–3D, respectively). VP1 is a structural and viral capsid protein of EV-A71. The main neutralizing antigenic determinants of the virus are concentrated on VP1, suggesting that VP1 has important research significance. Many studies have found that the mechanism by which EV-A71 regulates autophagy is related to VP1. In EV-A71-infected SK-N-SH cells, VP1 colocalizes with the autophagic marker protein LC3 [15]. Furthermore, replacement of EV-A71 VP1 decreases EV-A71 replication by regulating autophagy [16]. EV-A71 VP1 also promotes autophagy of Schwann cells in mice via endoplasmic reticulum stress upregulation [17]. However, the mechanism by which EV-A71 VP1 regulates autophagy is still unclear and needs to be investigated further.

Prohibitin 1 (PHB1) and PHB2 are two highly homologous members of the prohibitin (PHB) protein family. Prohibitins (PHBs) are ubiquitously expressed in the nucleus, mitochondria, and cytosol and play a role in transcription, nuclear signaling, and mitochondrial structural integrity [18]. Moreover, PHBs are reported to be involved in the infection process of viruses, including in HCV, DENV-2, and EV-A71 [19,20,21]. PHB2 acts as an autophagy receptor to induce autophagy [22,23]. These findings suggest a possible relationship between PHB2 and the autophagy mechanism during EV-A71 infection.

Virus-induced autophagosomes can fuse with lysosomes to form autolysosomes, which promote vesicle acidification and degradation by vacuolar ATPases [24], constituting complete autophagy. Viruses have developed strategies to impede fusion of autophagosomes with lysosomes to avoid autophagic degradation. Such virus-induced incomplete autophagy has been observed in CVB3-, rotavirus-, and influenza A virus (IVA)-infected cells [25,26,27]. Interestingly, studies have shown that EV-A71 forms autolysosomes in rhabdomyosarcoma (RD) cells [24]. Given that the acidic environment of autolysosomes should degrade the virus, this finding needs further examination.

In this study, we show that the C-terminus of VP1 is essential for induction of autophagy. We also demonstrate the interaction between PHB2 and EV-A71 VP1 by using mass spectrometry. Our results suggest that PHB2 is involved in EV-A71 infection via autophagy. Furthermore, we determine that EV-A71 induces complete autophagy that requires autolysosomal acidification, thus affecting EV-A71 infection. 

## 2. Materials and Methods 

### 2.1. Antibodies and Reagents

The following antibodies were used: rabbit anti-LC3B (Cell Signaling Technology, Boston, MA, USA, 3868), rabbit anti-PHB2 (Cell Signaling Technology, 14085), rabbit anti-SQSTM1/p62 (ABclonal Technology, Wuhan, China, A77580), normal rabbit IgG (ABclonal Technology, AC005), mouse anti-PHB2 (Proteintech, Wuhan, China, 66424-1-Ig), mouse anti-β-actin (Proteintech, 66009-1-Ig), rabbit anti-flag (Proteintech, 20543-1-AP), and rabbit anti-EV-A71 VP1 (Genetex, Irvine, CA, USA, GTX132339).

The following reagents were used: 3-methyladenine (3-MA; Selleck, Shanghai, China, S2767), rapamycin (Selleck, S1039), nontargeting siRNA (SiNC) and siRNA targeting PHB2 (SiPHB2) and ATP6AP2 (SiATP6AP2) (RiboBio, Guangzhou, China), Lipofectamine 3000 Reagent (Invitrogen, Carlsbad, CA, USA, L3000015), Lipofectamine RNAiMAX Reagent (Invitrogen, 13778-150), and primers for PCR and real-time quantitative PCR (RT-qPCR) (Invitrogen).

### 2.2. Cell Culture, Virus Isolation, and Titer Determination

Human rhabdomyosarcoma (RD) cells and HEK293T cells were cultured in Dulbecco’s modified Eagle’s medium (DMEM) containing 10% fetal bovine serum (FBS; Gibco, Gaithersburg, MD, USA) at 37 °C in 5% CO_2_.

The full-length infectious clone of EV-A71 (C4), which was used for construction of protein expression vectors, was kindly provided by Prof. Qin Chengfeng (Academy of Military Medical Science, Beijing, China). RD cells were infected with viruses, and the culture supernatant was harvested according to the cytopathic effect (CPE). Viruses were stored at 80 °C, and viral titers were determined in RD cells by measuring the median tissue culture infectious dose (TCID50).

### 2.3. Construction of the Protein Expression Vector

Full-length VP1 was equally divided into six segments (aa 1–50, aa 51–100, aa 101–150, aa 151–200, aa 201–250, aa 251–297). The full-length VP1 and VP1 gene fragments were amplified from the full-length infectious clone of EV-A71 (C4) using the corresponding forward and reverse primers (Appendix A). The sequences were cloned into the corresponding restriction sites (EcoRI and BamHI) of the plasmid pEGFP-N1 after double digestion, and successful construction of the protein expression vector was verified by DNA sequencing.

### 2.4. DNA and siRNA Transfection 

Cells (5 × 10^5^ cells/well) were plated in a 6-well plate. The next day, plasmid DNA or siRNA was transfected into the cells using Lipofectamine 3000 and Lipofectamine RNAiMAX reagents (Invitrogen) according to the manufacturer’s recommendations. RD cells were infected with EV-A71 after siRNA transfection for 24 h. The cells were pretreated with 3-MA (10 mM) for one hour before DNA transfection and with rapamycin (100 nM) for one hour before EV-A71 infection. The sequence of the siRNA specific for PHB2 was 5′-CCAAGGACTTCAGCCTCAT-3′. The sequence of the siRNA specific for ATP6AP2 was 5′-CTCTGTGAAAGAAGACCTT-3′.

### 2.5. RT-qPCR

Total RNA was extracted using TRIzol (TAKARA, Dalian, China) at different time points after viral infection. Then, 1 μg of total RNA was transcribed to cDNA using a PrimeScript RT Reagent Kit with gDNA Eraser (TAKARA). Real-time quantitative PCR was performed using SYBR Premix Ex Taq (TAKARA) to detect GAPDH, PHB2, and ATP6AP2 mRNA or using Bestar qPCR MasterMix (DBI^®^ Bioscience, Ludwigshafen, Germany) to detect EV-A71 RNA. The specific primers and probes for EV-A71 and the PCR conditions have been described previously [28]. All primers used for RT-qPCR analysis are provided in Appendix A. The mRNA expression levels were normalized to those of GAPDH and are expressed as fold changes calculated using the 2^−ΔΔ*C*t^ method.

### 2.6. Western Blot Analysis

Cells were washed twice with 1× phosphate-buffered saline (PBS) after DNA transfection for 48 hours or at different time points after infection and then lysed with RIPA lysis buffer (Beyotime, Shanghai, China) on ice for 30 min. The supernatant was harvested after the cell lysates were centrifuged at 14,000 rpm for 30 min at 4 °C. After measurement of protein concentrations with a BCA protein assay kit (Bioworld, Minneapolis, MN, USA), 60 µg of protein per sample was separated by SDS-PAGE. The proteins were transferred to polyvinylidene difluoride (PVDF) membranes (Bio-Rad, Shanghai, China), which were blocked with 3% BSA in TBST for 2 h at room temperature and were incubated with the corresponding primary antibodies overnight at 4 °C. The next day, the membranes were incubated with secondary antibodies (Bioworld) for 1 h at room temperature, and proteins were detected with ECL reagent (Bioworld). The bands were quantified using the grey analysis method in ImageJ.

### 2.7. Immunofluorescence Analysis

At different time points after viral infection, RD cells were fixed with precooled methanol for 5 min and then permeabilized with 0.1% Triton X-100 for 5 min. The cells were blocked with 10% goat serum in PBST for 1 h, incubated with rabbit anti-EV-A71 VP1 and mouse anti-PHB2 antibodies overnight at 4 °C, and incubated with goat anti-rabbit-CoraLite 488 and goat anti-mouse-CoraLite 594 antibodies (Proteintech) for 1 h. Nuclei were stained with DAPI (BestBio, Shanghai, China). Images were acquired using an Olympus FluoView FV10i confocal microscope (Olympus, Tokyo, Japan) and analyzed with FV10-ASW 3.0 Viewer software (Olympus).

### 2.8. Coimmunoprecipitation (co-IP) Analysis

According to the protocol of the co-IP kit (Thermo Fisher, Rockford, IL, USA, 26149), the anti-flag antibody, anti-PHB2 antibody, and normal rabbit IgG were immobilized on the resin. Cell lysates were harvested with IP lysis/wash buffer and precleared using control agarose resin. The protein mixture was added to the appropriate antibody-coupled resin and incubated with gentle mixing overnight at 4 °C. The next day, the coimmunoprecipitates were eluted, and samples were prepared for SDS-PAGE.

### 2.9. TCID50 Assay

Supernatants were collected at different time points after infection for viral titer determination. RD cells (10^4^ cells/well) were plated in a 96-well plate. The next day, the cells were incubated with tenfold dilutions of infected-cell supernatants for 1 h, after which the cells were washed with PBS once and cultured in DMEM containing 2% FBS.

On day four or five after infection, the number of plates exhibiting the CPE was recorded to determine the viral titer using the Reed–Muench method.

### 2.10. CCK-8 Assay

RD cells (10^4^ cells/well) were plated in a 96-well plate, transfected with siRNA for 24 h, and then infected with EV-A71 for 48 h. Then, 0.1 volume of CCK-8 solution (DOJINDO, Shanghai, China) was added to each well and incubated for 4 hours at 37 °C. The absorbance of each well at 450 nm was measured using a microplate reader (Tecan Infinite M200, Tecan Group Ltd., Männedorf, Switzerland). The survival rate of uninfected cells was set to 1, and the survival rates of the other groups were calculated.

### 2.11. Measurement of Lysosomal pH

A standard pH curve was generated as follows. Resuspended cells were incubated with probes diluted in HBSS for 30 min in the dark. After centrifugation, the cells were resuspended again with the pH standard solutions (135 mM KCl, 2 mM K_2_HPO_2_, 20 mM HEPES, 1.2 mM CaCl_2_, and 0.8 mM MgSO_4_; pH 4–6.5), and 50 µM nigericin was added to the cells and incubated for 10 min. The fluorescence intensity was measured using a microplate reader (excitation/emission = 385/440 nm (F1) and 385/540 nm (F2)). The standard curve was plotted in Microsoft Excel software with the pH values in buffer solution on the horizontal axis and the F2/F1 ratio on the vertical axis.

RD cells were plated in a 96-well plate and transfected with SiNC- or siRNA-targeting ATPAP2. At the corresponding time point, the medium was replaced with HBSS buffer containing 1000-fold diluted acidophilic fluorescent probes (BBcellProbe^®^P02, BestBio). The cells were incubated for 5 min at 37 °C, and the fluorescence intensity was measured immediately using a microplate reader (excitation/emission = 385/440 nm (F1) and 385/540 nm (F2)). The pH values were calculated according to the standard pH curve. 

### 2.12. Statistical Analysis

Data from three independent experiments are presented as means ± SEMs determined using GraphPad Prism 7. The significance of differences was analyzed by one-way ANOVA or Student’s t test in SPSS 24.0. A *p* < 0.05 was considered to indicate a significant difference.

## 3. Results

### 3.1. The C-Terminus (aa 251–297) of VP1 Is Required for Induction of Autophagy

We constructed pEGFP-N1 expression vectors expressing full-length VP1 protein and VP1 fragments with flag tags (VP1-flag, VP1 (aa)-flag). The flag tag proteins were detected at the correct molecular weight (Figure 1A), indicating that the expression vectors were successfully constructed. After the VP1-flag was transfected into HEK293T cells, the LC3-II/LC3-I ratio in the VP1-flag group was significantly higher than that in the mock and flag groups, and this effect was inhibited by 3-MA (Figure 1B), a selective PI3K inhibitor that blocks early autophagosome formation [29]. These data suggest that the VP1 protein induces autophagy. To further explore the site responsible for VP1-induced autophagy, we divided VP1 into 6 fragments and constructed 6 expression vectors with flag tags. When autophagy occurs, the microtubule-associated protein light chain 3-I (LC3-I) in the cytoplasm is recruited to the autophagosome membrane and modified to form lipophilic LC3 (LC3-II). The degree of LC3-I-to-LC3-II conversion is usually used as an indicator of autophagosome formation [30,31]. After these expression vectors were transfected into HEK293T cells, the LC3-II/LC3-I ratio in the VP1 (aa 251–297)-flag group was significantly higher than that in the other VP1 fragment groups (Figure 1C) and was closest to that in the VP1-flag group. These results indicate that the C-terminus (aa 251–297) of VP1 is required for induction of autophagy.

### 3.2. EV-A71 VP1 Interacts with PHB2

To further study the mechanism by which the EV-A71 VP1 protein regulates autophagy, we searched for host proteins involved in this mechanism. Because the site of VP1 responsible for autophagy induction is located at the C-terminus (aa 251–297), we transfected VP1 (aa 1–250)-flag, VP1 (aa 251–297)-flag, and VP1-flag into HEK293T cells and subjected the transfected cells to co-IP using anti-flag antibodies. Differential bands were found at approximately 35 kDa after silver staining (Figure 2A). Mass spectrometry was performed on the excised gel strips to screen for proteins that might interact with the VP1 protein. To identify autophagy-related proteins that interact with VP1, we used band 1 as the control and narrowed down the final range to the proteins contained in band 2 or 3 but not in band 1. The lists of proteins contained in bands 1, 2, and 3 of Figure 2A correspond to Appendix A, Appendix A, and Appendix A, respectively. Based on the molecular weight of the bands (~35 kDa), the candidate proteins were the following: PHB2, RPS4X, RPS3A, RPL7A, SLC25A6, and PRPS1. RPS4X, RPS3A, and RPL7A are components of the ribosome and participate in translation. SLC25A6 functions as a gated pore that translocates ADP and ATP. PRPS1 is an enzyme necessary for nucleotide biosynthesis. Among these proteins, PHB2 is the only one related to autophagy [22,23]. 

We selected PHB2 as the protein of interest for further analysis. PHB2 has a molecular weight of 33 kDa and is an autophagy-related protein. VP1-flag was transfected into HEK293T cells, and co-IP results showed that PHB2 interacted with VP1 (Figure 2B). The co-IP results in EV-A71-infected RD cells also indicated this interaction (Figure 2C). According to immunofluorescence assays, the VP1 and PHB2 proteins colocalized at various time points after infection (Figure 2D). The above results indicated that PHB2 and VP1 interact, and we speculated that this interaction is related to autophagy.

### 3.3. PHB2 Is Involved in EV-A71 Infection via Autophagy

To further study the role of PHB2 in EV-A71 infection, siRNA targeting PHB2 was used to silence PHB2. SiPHB2 was transfected into RD cells prior to infection, and good knockdown efficiency (>80%) was achieved at the indicated time points after infection (Figure 3B,C). PHB1, the other highly homologous PHB, is reported to be involved in EV-A71 infection [24]; thus, it was necessary to rule out the possibility of PHB1 downregulation along with PHB2 knockdown and subsequent PHB upregulation due to EV-A71 infection, and we did not observe down- or upregulation of PHB1 after PHB2 knockdown and EV-A71 infection (Figure 3C). Knockdown of PHB2 resulted in an increased cell survival rate (Figure 3A). At various time points after infection, the intracellular viral RNA in the SiPHB2 group was significantly reduced compared to that in the SiNC group (Figure 3D), and viral titers in the supernatants of infected cells showed a trend consistent with this reduction (Figure 3E). These results indicate that the replication and release of EV-A71 were inhibited after knockdown of PHB2. Knockdown of PHB2 also resulted in a significant decrease in the LC3-II/LC3-I ratio, indicating that autophagy was inhibited. Moreover, VP1 protein expression was significantly reduced (Figure 3F). Rapamycin is a specific mTOR inhibitor that induces autophagy [32], and rapamycin treatment significantly promoted autophagy (LC3-II/LC3-I ratio) and VP1 expression (Appendix A). We also found that rapamycin treatment of PHB2-knockdown cells markedly reversed PHB2 knockdown-mediated inhibition of autophagy (LC3-II/LC3-I ratio) and VP1 protein expression. Interestingly, compared to the results illustrated in Appendix A, this increase only reached the level close to that of the control group, indicating that the knockdown of PHB2 also inhibited the autophagy induced by rapamycin treatment and ensuing VP1 expression. Taken together, these results demonstrate that PHB2 is involved in EV-A71 infection via autophagy.

### 3.4. EV-A71 Induces Complete Autophagy that Requires Autolysosomal Acidification, thus Affecting EV-A71 Infection

To explore the relationship between autophagy and EV-A71 infection, we investigated whether autolysosomal acidification is required for EV-A71-induced complete autophagy and EV-A71 infection. The vacuolar ATPase (V-ATPase) complex is an ATP-dependent proton pump mainly responsible for pumping protons (H^+^) into the lysosome to acidify it [33], and deletion of the V-ATPase subunit inhibits lysosomal acidification without disrupting autophagosome–lysosome fusion [34]. Among the many subunits of the V-ATPase complex, we selected ATP6AP2 (SiATP6AP2) for V-ATPase knockdown. ATP6AP2 is an important auxiliary component of the V-ATPase complex and coordinates correct V-ATPase assembly [35]. To evaluate the effect of ATP6AP2 knockdown on lysosomal pH, we employed an acidophilic fluorescent probe to measure the lysosomal pH, which increased significantly, as expected (Figure 4A). The knockdown efficiency of ATP6AP2 was almost 90% (Figure 4C,F), and the cell survival rate increased after knockdown (Figure 4B). Knockdown of ATP6AP2 resulted in a reduction in intracellular viral RNA titers (Figure 4D), and viral titers in the supernatant also decreased with the same trend (Figure 4E). The LC3-II/LC3-I ratio and SQSTM1/p62 protein (an autophagy degradation substrate) expression in the SiATP6AP2 group increased, indicating that autolysosomal degradation was inhibited. In addition, VP1 protein expression was significantly reduced (Figure 4F). These results suggest that EV-A71 infection is partly dependent on the acidic environment of autolysosomes; in other words, EV-A71 induces complete autophagy that requires autolysosomal acidification, thus affecting EV-A71 infection.

## 4. Discussion

Recently, an increasing number of studies have shown that viruses can manipulate autophagy for their own benefit. Viruses often regulate autophagy via host protein–virus interactions. The M protein of human parainfluenza virus type 3 (HPIV3) interacts with the TUFM protein to regulate the formation of autophagic vesicles, and its P protein can interact with SNAP29 and competitively inhibit the interaction with STC17, thereby preventing the fusion of autophagosomes and lysosomes [36]. In addition, the 2BC protein of EV-A71 interacts with the SNARE protein to form autolysosomes [24]. Such interactions between viral and host proteins led us to investigate the mechanism by which EV-A71 induces autophagy, and our study of this mechanism focuses on the interaction between viral and host proteins.

We first verified that the EV-A71 VP1 protein induces autophagy and determined that the site of VP1 responsible for autophagy induction is located at the C-terminus (aa 251–297). Next, we used co-IP and mass spectrometry to study the interactions and screened the results to identify a protein of interest, PHB2. After verification experiments, interaction between VP1 and PHB2 was finally confirmed. PHB2, known as repressor of estrogen receptor activity (REA) or B-cell receptor associated protein (BAP)-37, is one of two PHB proteins encoded in the human genome, along with prohibitin 1 (PHB1) [37,38]. PHBs are found in the nucleus, mitochondria, and cytosol [18]. Studies have shown that PHB2 acts as an autophagy receptor to induce this process [22]. PHB2 may interact with EV-A71 VP1 to regulate autophagy, thus affecting EV-A71 infection.

Through various experimental methods, we demonstrated that PHB2 participates in EV-A71 infection by regulating autophagy. After PHB2 knockdown, viral replication and release were inhibited, indicating that EV-A71 replication and viral particle release are dependent on PHB2. In addition, autophagy was inhibited and VP1 protein expression was reduced after PHB2 knockdown. Interestingly, knockdown of PHB2 also inhibited the autophagy induced by rapamycin treatment as well as VP1 expression, further indicating that PHB2 can participate in the process of EV-A71 infection by regulating autophagy. PHB1 and PHB2 usually exist as heterodimers in mitochondria [39]. HIV replication relies on binding of the PHB1–PHB2 complex to the C-terminus of HIV glycoproteins [40]. Studies have found that cell surface-expressed PHB1 can act as a receptor for EV-A71 entry into NSC-34 cells and that intracellular (mitochondrial) PHB1 is involved in EV-A71 replication [21]. These findings indicate that PHB2 is likely to be involved in EV-A71 infection. Our study demonstrated this involvement and found that it is mediated via autophagy. The inner mitochondrial membrane protein PHB2 can induce mitophagy by binding to the autophagosome membrane protein LC3 [22,23]. However, interestingly, we did not find colocalization of or interactions between PHB2 and LC3 in EV-A71-infected cells (data not shown), suggesting that EV-A71 does not induce mitophagy. Accordingly, rather than by directly binding to LC3, PHB2 may regulate autophagy during EV-A71 infection through mechanisms that need further investigation.

PHBs are associated with certain cell membrane receptors. The mechanism by which PHB1 acts as a membrane receptor mediating EV-A71 entry into NSC-34 cells has not been found in RD cells [21]. In RD cells, SCARB-2 is the main receptor for EV-A71 entry, and it does not mediate EV-A71 entry into NSC-34 cells [41]. Given the differences between nerve cells and muscle cells, PHB2 is unlikely to mediate EV-A71 entry into RD cells. Moreover, we did not observe an interaction between VP1 and PHB1 (data not shown) or an effect of PHB2 knockdown on PHB1 expression. These data suggest that the mechanism we describe for PHB2 is independent of PHB1.

We further investigated EV-A71-induced complete autophagy. EV-A71 is reported to induce the formation of autolysosomes, and treatment with lysosomal acidification inhibitors (chloroquine, bafilomycin A1, and NH_4_Cl) inhibits EV-A71 infection [24]. However, this is counterintuitive, as lysosomal acidification (pH 4.2–5.3) enables optimal activity of various lysosomal hydrolases, which should facilitate viral degradation rather than replication. These inhibitors have two separable and independent effects. In addition to directly acting on V-ATPase to inhibit lysosomal acidification, lysosomal acidification inhibitors can disrupt autophagosome–lysosome fusion in a manner independent of V-ATPase [34]. In fact, deletion of the V-ATPase subunit can inhibit lysosomal acidification without disrupting autophagosome–lysosome fusion [34]. Consequently, we chose to knock down the key component of V-ATPase, ATP6AP2, to assess EV-A71 infection. Our results show reduced EV-A71 replication, viral particle release, and viral protein synthesis after knockdown. These data suggest that EV-A71 infection partly depends on autolysosomal acidification, which further strengthens the relationship between autophagy and EV-A71 infection. Notably, DENV, PV, and HCV, which are also positive-strand RNA viruses, can form autolysosomes to facilitate their infection [42,43,44,45]. The complex relationship between viruses and autolysosomes deserves further study.

In conclusion, we found that the host protein PHB2 is involved in EV-A71 infection and enhances the autophagy mechanism during EV-A71 infection. PHB2 may be a novel host target for anti-EV-A71 drug development.

## Figures and Tables

**Figure 1 viruses-12-00414-f001:**
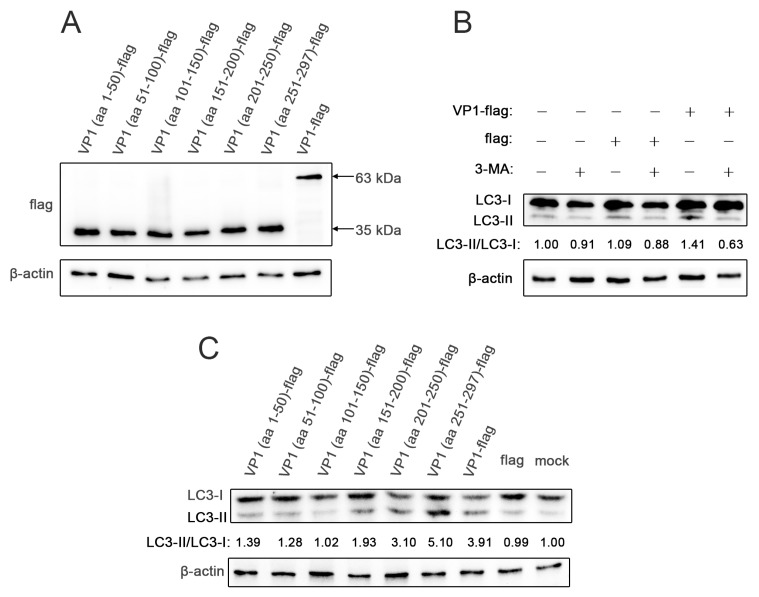
Enterovirus (EV)-A71 VP1 induces autophagy via residues 251–297. (**A**) Vectors expressing six VP1 fragments and full-length VP1 were transfected into HEK293T cells, and proteins were detected by Western blot analysis. (**B**) VP1-flag and flag were transfected into HEK293T cells pretreated with or without 3-MA, and proteins were detected by Western blot analysis. (**C**) HEK293T cells were transfected with different expression vectors, and proteins were detected by Western blot analysis.

**Figure 2 viruses-12-00414-f002:**
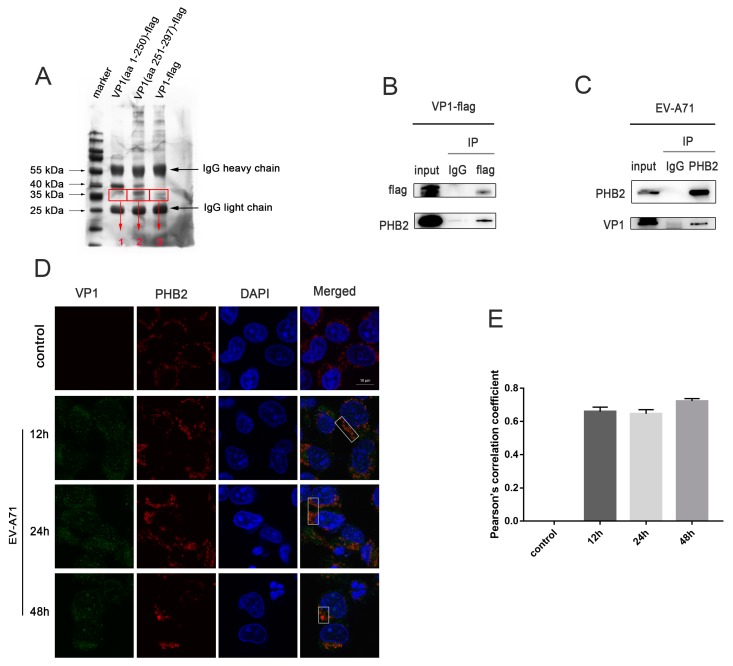
The EV-A71 VP1 protein interacts with PHB2. (**A**) HEK293T cells were transfected with VP1 (aa 1–250)-flag, VP1 (aa 251–297)-flag, and VP1-flag, and cell lysates were subjected to a co-IP assay with an anti-flag antibody. The coimmunoprecipitated proteins were separated by SDS-PAGE and visualized by silver staining. Three different gel bands were sent for mass spectrometry identification after tryptic in-gel digestion. The lists of proteins contained in band 1, 2, and 3 of Figure 2A correspond to Appendix A, Appendix A, and Appendix A, respectively. (**B**) VP1-flag was transfected into HEK293T cells, and cell lysates were subjected to a co-IP assay with an anti-flag antibody or with normal IgG as a negative control. The final coimmunoprecipitated proteins were detected by Western blot analysis. (**C**) rhabdomyosarcoma (RD) cells were infected with EV-A71 (multiplicity of infection (MOI) = 1) for 24 h, and cell lysates were subjected to a co-IP assay with an anti-PHB2 antibody or with normal IgG as a negative control. The final coimmunoprecipitated proteins were detected by Western blot analysis. (**D**) RD cells were infected with EV-A71 (MOI = 1), fixed at several time points post infection, and stained with antibodies against VP1 (green) and PHB2 (red). DAPI staining is shown in blue. Scale bars = 10 μm. (**E**) Pearson’s correlation coefficient (PCC) was calculated from 3 regions of interest (ROI) in each graph.

**Figure 3 viruses-12-00414-f003:**
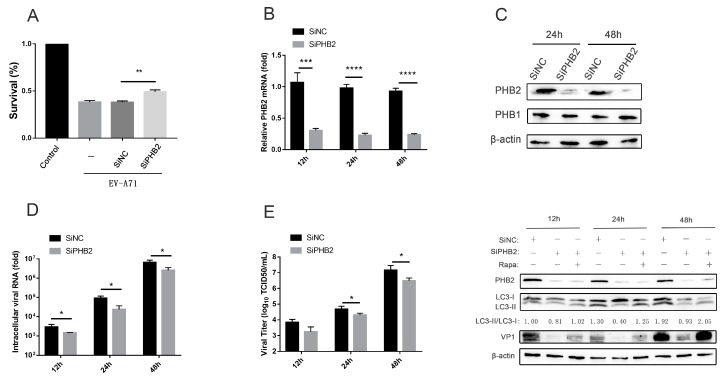
Knockdown of PHB2 inhibits EV-A71 infection and autophagy. (**A**) RD cells were transfected with SiNC or SiPHB2 for 24 h and then infected with EV-A71 (MOI = 5) for 24 h. The cell survival rate was determined by a CCK-8 assay. (**B**–**E**) RD cells were transfected with SiNC or SiPHB2 for 24 h and then infected with EV-A71 (MOI = 1). At different time points after infection, PHB2 mRNA (**B**) and intracellular viral RNA (**D**) levels were measured via RT-qPCR, (E) viral titers in the supernatants of infected cells were determined by TCID50 assays, (**C**) and (**F**) proteins were detected by Western blot analysis. * *p* < 0.05, ** *p* < 0.01, *** *p* < 0.001, and **** *p* < 0.0001.

**Figure 4 viruses-12-00414-f004:**
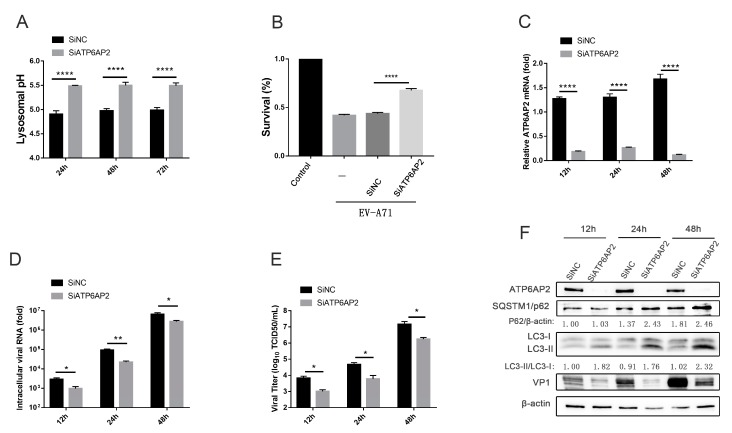
Knockdown of ATP6AP2 (V-ATPase) suppresses EV-A71 infection. (**A**) RD cells were transfected with SiNC or SiATP6AP2, and the lysosomal pH was measured using fluorescent probes at different time points. (**B**) RD cells were transfected with SiNC or SiATP6AP2 for 24 h and then infected with EV-A71 (MOI = 5) for 24 h. A CCK-8 assay was performed to determine the cell survival rate. (**C**–**F**) RD cells were transfected with SiNC or SiATP6AP2 for 24 h and then infected with EV-A71 (MOI = 1). At different time points after infection, ATP6AP2 mRNA (**C**) and intracellular viral RNA (**D**) levels were measured via RT-qPCR, (**E**) viral titers in the supernatants of infected cells were determined by TCID50 assays, and (**F**) proteins were detected by Western blot analysis. * *p* < 0.05, ** *p* < 0.01, and **** *p* < 0.0001.

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
