# Peer review of "Interaction between PHB2 and Enterovirus A71 VP1 Induces Autophagy and Affects EV-A71 Infection"

_viruses, 2020, doi:10.3390/v12040414_

Round 1

Reviewer 1 Report

The approach and logic are

On page 9 line 40: Please change we further studied ...  to we " have studied"...

page 9 Line 343: Our study found ... to "In our study we showed".... 

I think few lines about prohibitin (PHB) and it's importance should be discussed in the discussion, or few lines on PHB2 can be added to the Introduction 

Reviewer 2 Report

The Interaction between PHB2 and EV71 VP1 is 3 Involved in Enterovirus 71 Infection via Autophagy

The manuscript presents data originated to show the involvement of PHB2 in EN71 replication process via VP1 protein enhancing autophagy mechanisms to increase virus yields in a cell model system. The main concern is the overall effect of PHB2 in virus replication and

The involvement of prohibitin in EV71 entry-replication has been published in 2018 by Issac Horng Khit Too (doi: 10.1371/journal.ppat.1006778) showing the interaction of the protein in EV71 infected cells by using cell lines knocked down for PHB (another highly homologous members of this family). The article demonstrates the interaction of the VP1 of EV71 with the surface expressed prohibitin.

Would this be a possibility that PHB2 also interacts with the VP1 during virus entry? If so, how you can distinguish between the published paper and your report?

Figure 1 shows LC3-I and LC3-II in panel “A”. the bands should be quantified and normalised to beta actin and then presented in a form of table or graph to represent the differences between different constructs and treatments.

Part of the reasoning that AA251-297 interacts with PHB2 comes from figure 1 panel “C”. This part is elusive since the other constructs also interact and induce LC3-II. The same issue of quantification and normalizing the intensity of the band is needed to represent the data.

Would it be possible to rename the constructs with the AA numbers rather than the nucleotide numbers? This would be easier to follow.

Why did you not investigate other VP or non-structural proteins of the EV71 and only focused on the VP1? The reasoning behind the VP1 investigation has not presented here.

In figure 2 panel G the colocalization is not very convincing, however; the most detected PHB2 detected is limited to VP1 expression sites. This is not enough to convincingly present the colocalization though.

In figure 3 the survival of the cells in SiPHB2 transfected RD cells is slightly higher than non-transfected or sham-transfected cells (- or SiNC) even though the expression of the mRNA of PHB2 has reduced dramatically (panel B figure 3). Also, the reduction in the virus mRNA yields in SiPHB2 transfected compared to control (SiNC) is modest and do not support the SHP2 having a dramatic effect in virus replication when blocked in this setting. Although you have shown the reduction is significant (in panel C) this reduction is less than one log therefore I am not very much convinced that the PHB2 plays a key role in EV71 replication process at this setting. Also, the virus titters are reduced less than one log in time course analysis (shown in panel D figure 3). However, the virus capsid protein VP1 shows more accumulation in controls than rapamycin treated SiPHB2 transfected RD cells. This points out that the lack of PHB2 due to reduction of the PHB2 mRNA has almost no effect on VP1 production. This means that there must be another mechanism to reduce the virus replication (even though it is less than a log reduction) in this experiment. This finding contradicts the findings that PHB2 plays a role in EV71 replication and you need to explain this in more details.

For the same reasoning the virus yields do not decrease dramatically (less than one log) in the ATP6AP2 (v-ATPase) knockdown experiments presented in figure 4. Therefore, the role of acidic environment of the autolysosome in the rate of EV71 replication is not huge rather a minor effect.

Increasing the survival of the cell is different than virus replication rate and showing the increase in the survival rate of the cell by blocking the autophagy mechanisms presumably by PHB2 mechanism do not represent a great impact on enhancing EV71 replication. There might be other mechanisms that PHB2 is involved that might affect the EV71 replication such as virus entry as shown for PHB1. You have not formally ruled out the possibility of degrading the PHB1 or down regulation along with PHB2 knock down. Alternatively, the absence of PHB2 could increase the bioavailability of the PBH1 and enhance the virus spread, therefore, showing a minor effect.

All in all, I am not convinced that PHB2 is a key player in EV71 replication process in the way that authors have drafted. More evidence is required to merit this manuscript publication consideration.

Reviewer 3 Report

Thank you for the interesting paper. I have it through carefully, and I have made few remarks that I would like you to address in your revised version of the paper.

Starting with the title. Emphasis seems to be in “interaction – involved”. I would propose modification of the title, e.g. interaction – facilitates, or Role of X and Y interaction during EV-A71 infection via autophagy. You could also remove “The” as “the” and “a” are not usually used in the title.

Check the nomenclature for the name of the virus type (enterovirus A71). See Simmonds et al. 2020. Recommendations for the nomenclature of enteroviruses and rhinoviruses. Archives of Virology. https://doi.org/10.1007/s00705-019-04520-6. Follow these guidelines throughout the text.

Page 1, line 9: HFMD is not a disease that endangers lives of children. It is a disease described by fever and spots. EV-A71 may cause CNS disease, which may be lethal. That is different from HFMD! Check line 32.

line 10: Autophagy is an important mechanism in the response to pathogen infection, but many studies have shown that RNA viruses can utilize autophagy to promote self-replication. Why “but”? Please, explain.

line 12: EV71 infection is closely related to its autophagy induction mechanism. I do not understand the relation between virus infection and autophagy. Please, explain.

line 15: we first found that VP1 induces autophagy via a site at the carboxyl terminus (aa 251-291). You probably mean that induction of autophagy is mediated by interaction of VP1 C-terminus with…., please, modify accordingly.

line 16: found by mass spectrometry that the host protein prohibitin 2 (PHB2) may interact with VP1 and then verified this interaction. This would run better, if it reads as: MS analysis suggested that PHB2 interacts with C-terminus of EV-A71 VP1 protein. This was further verified by PHB2 knock-down….

line 18: These results suggested that PHB2 interacts with VP1 to participate in EV71 infection via autophagy. You probably mean that the results suggested that PHB2 interaction with VP1 is essential for induction of autophagy and infectivity of EV-A71.

line 22: knowledge that will guide the identification of host targets for anti-EV71 drugs. There is no information to corroborate the information in this sentence. Please, remove it, and do not repeat elsewhere. The importance of the findings must lie elsewhere.

To summarize the abstract, basically, every sentence in it needs polishing. I will strongly recommend proof-reading of the paper before re-submission (or actually it is a must!).

line 34: I think it is about the efficacy of the vaccine. I do not see place for this sentence in this position of the text. Please, remove.

line 36: Please, remove.

line 50: EV-A71 specs in separate chapter, or make it very short and link preceding autophagy to VP1 and autophagy leaving few facts about the virus in between (see the following comment):

line 54-56: remove these sentences and bridge VP1 directly to autophagy.

lines 57-59: Sentences could be combined… and/while… but these could be placed in the following chapter.

line 59: Again a non-sense sentences. Do not make statements that may not be valid (unless you go through literature).

lines 62-69: why do you hop in and out between lysosomes and exit – they are not the same thing. Explain one first, then the other.

line 98: Please, elaborate the fragments, and not just point to Table S1. Later on it is evident that the reader gets confused by numbering of the constructs, whether nucleotides or amino acids.

line 177: Check Figure 1. I cannot follow the text and figure captions. Are the figures in wrong order or what?

line 184: If written like this, indicate the name in parenthesis after the name of the construct, that is, pEGFP-N1 expression vector expressing the VP1 protein with a flag tag

line 185: This effect was reversed by…

line 186: These data suggest that..

line 193: Is VP1 (aa 291-297) really in the C-terminus? Or, is the numbering in Fig 1 in nucleotides, and if so, why?

Overall, I do not see real effect by looking at the figures. The bands must be quantified.

line 202: We preferentially considered…. I do not understand what is meant by this sentence. Please, elaborate.

line 207: Place the hit list as supplement and elaborate the content in the text.

line 214: The possible co-localization is beyond my vision or imagination. The images must be processed by a program that allows black-white resolution of pixels to verify co-localization (costes p-value calculations).

line 232: Is there a visible effect in infectivity in cells that could be visualized by fluorescence microscopy? If virus replication is really inhibited at the level of VP1 expression, this might be visible as virus not being visible. However, the overall effect on infectivity, despite p-value calculations, is not impressive. This is a bit confusing at this point.

line 234: Where is the data for cell fate?

line 235: Survival in which conditions? This is not very clear.

line 242: Rapamycin ,where, how? IF rapamycin inhibits autophagy, how does that relate to virus replication, VP1 production, PHB2 etc. Why was the experiment done?

line 255. I do not see how this part supports the overall study related to VP1 and PHB2.

I did not even bother to look at the discussion, since it will need modification once the preceding writing is corrected.

Round 2

Reviewer 2 Report

Thank you for the significant improvement of the manuscript.

There are two minor points that need to be addressed.

In figure 1B&C the ratio between LC3-II and LC3-1 do not seem to be correct since the bands correlating the LC3-II are weaker than of LC3-I. This needs to be explained. The quantification method of the bands has not been explained.

The mass spectrometery data have been transferred to Supplementary material but the marks in Figure 2A have been left without explanation in the figure legend. Could you please add the explanation to the figure legend and refer to the supplementary material for clarity.

Also the pictures of the colocalizations are still hazy but the graphs makes it more plausible.

Reviewer 3 Report

I find the data interesting but there are few points that need to be addressed. I find the paper lacking proof-reading and difficult to read. The data shows effect of VP1 on autophagy (Fig 1) and interaction VP1 with PHB2 (Fig 2), but it fails to demonstrate the relation between the effect of PHB2 knock-down and VP1 in virus infection. There is a critical control missing, that is, virus lacking VP1 or preferably the interacting motif. This should be compared to native virus infection.

Other remarks below:

Title: I think it would be better to write: … PHB2 and enterovirus A71 is involved in virus infection via autophagy. That is, write full virus name first, and secodn do not repeat it later.

Remove: “The”

Having said that, I do not think that “interaction” and “involved” makes a good sentence. Maybe “play a role”, but how about: Interaction between PHB2 and enterovirus A71 VP1 induces (or enhances) autophagy and affects EV-A71 infection

Line 16: remove “important”

Line 18: first,…. and where is the second??? Bad English.

Line 21: and this further verified… and missing something. Bad English.

Line 26: What is the relation of this part to the work?

Line 36: Large-scale outbreaks vs. varying degrees. Bad English.

Line 37: children under 5 years of ago. Bad English.

Line 42-44: BAD, BAD English. These are nonsense declarations and have nothing to do with your study.

Line 51: coordination among proteins. Bad English.

Line 52: You missed to tell what is autophagosome. Despite… is useless here. Remove!

Line 56-59. Cannot read this sentence. Bad English.

Line 80. If you open up PHB as prohibitin, it should be self-evident that you write: PHB1 (prohibitin 1) and PHB2….

Line 80: What does “pair of proteins” mean? Dimer? Each protein subunit has a role in autophagy or infection? See below:

Line 83: Go through the text and modify all places where you use “can” to PHB2 acts as….

Line 82-84: If PHB1 is involved in infection process and PHB2 is autophagy receptor, how come PHB2 could show relation to autophagy and infection? Are you addressing PHB1 or PHB2, or both?

Line 85: Might be interesting but not in this paper. It is not very scientific to make statements in the middle of sentences about something that might be interesting. Focus!

Line 86: You already said that (line 52).

Line 90: What is “incomplete” autophagy? You must first explain “autophagic degradation” on line 85.

Line 90: IVA????

Line 93: Where do you explain autolysosome?

Line 93-94: I do not think you can deduce the relation between replication and induction of autolysosomes without explaining what autolysosomes are including pH.

Line 97: In this study, we show that VP1 C-terminus is essential for induction of autophagy… We also demonstrate the interaction…. by using MS.

Line 99. These data suggest that…..

Line 100-101: I do not see the relation of this to the overall story. It does not improve your story unless you somewhat tie it to the autophagy induction or infectivity. Sometimes less is more…..

Line 119: Why cDNA clone? Explain where it was used here even if it was told in the next section.

Line 150: Cells were washed… after DNA…

Line 207: C terminus of VP1 is required for induction or essential in….

Line 214 and 218: MA role should be placed after sentence on line 2018.

Line 215: Following the transfection of….

Line 217: Should it be “prevented” and not “reversed” since in the presence of MA autophagy should not happen?

I do not see the difference. There should be a marker that shows a real conversion of LC I to LC II, and VP1 effect should be compared to that. If one compares to VP1 (aa) flag constructs, why do I not see similar effect in 1B compared to 1C???

Line 242-248. The list of irrelevant proteins should be shown in the supplement and only show the sentence in the line 248

Line 249: Why “finally”?

Line 250: What “reasoning”?

Line 258: Please, use the same names for the constructs as in Fig 1. This is just confusing the reader.

Line 259: After “the final coimm….proteins were….”???

Line 271: You have missed the calculations for the control. Take any region and assume….

LIne 287: You should show the effect of rapa to untreated cells without SiPHB2 and demonstrate that the presence of PHB2 affect autophagy in the presence of rapa. That is, which one is stronger.

Line 303: I do not find any reason why this result is in this paper and what is its relation to PHB2 and VP1 and autophagy. Do you?

Round 3

Reviewer 3 Report

I wish to thank you for prompt responses to the reviewer criticism. I find in most part those satisfactory.